# The Association Between Obstructive Sleep Apnea and Oral Function, Using the Korea National Health and Nutrition Examination Survey Data

**DOI:** 10.3390/healthcare13111323

**Published:** 2025-06-02

**Authors:** Keon Woo, Junghoon Lee, Chae-Eun Jung, Jungwon Park, Yoonsoo Choy

**Affiliations:** 1Graduate School of Transdisciplinary Health Sciences, Yonsei University, Seoul 03722, Republic of Korea; wwhh10@yuhs.ac; 2Department of Smart Healthcare Information, Healthcare Management, Eulji University, Seongnam 13135, Republic of Korea; tmvlflt124@naver.com (J.L.); jce3511@naver.com (C.-E.J.); djwm3@naver.com (J.P.)

**Keywords:** obstructive sleep apnea, oral function, socioeconomically disadvantaged groups

## Abstract

**Background**: Obstructive sleep apnea (OSA) has been associated with adverse oral function outcomes, yet its association with oral function remains underexplored. This study aims to analyze the association between OSA and oral function problems, and whether these relationships are more pronounced among socioeconomically disadvantaged groups. **Methods**: Data were derived from the 2022–2023 Korea National Health and Nutrition Examination Survey (KNHANES), including 6349 participants aged 40 and above. OSA risk was assessed using the STOP-Bang questionnaire. Oral function was evaluated through chewing discomfort, speaking discomfort, and dental pain. Complex sample logistic regression was performed, adjusting for demographic, socioeconomic, and health-related covariates. Stratified analyses were conducted to examine whether the association differed across socioeconomic groups. **Result**: High OSA risk was significantly associated with chewing discomfort (OR: 1.365, 95% CI: 1.121–1.662), speaking discomfort (OR: 1.534, 95% CI: 1.126–2.082), and dental pain (OR: 1.198, 95% CI: 1.006–1.431). Stratified analyses showed stronger associations in low education or income groups. For instance, those in the lowest income group were over five times more likely to report speaking discomfort (OR = 5.207, 95% CI: 2.365–11.462) than those in the highest. **Conclusions**: OSA risk is significantly associated with impaired oral function, particularly among socioeconomically disadvantaged groups. These findings underscore the need for integrated public health approaches that address both sleep and oral health disparities.

## 1. Introduction

Sleep is a fundamental physiological process crucial for human health, impacting the quality of life through its restorative functions, including energy regulation, memory consolidation, and immune system support [1]. Adequately, high-quality sleep is, thus, indispensable for overall well-being and productivity. However, a concerning global trend shows a decline in adult sleep duration, exacerbated by factors associated with economic development and evolving societal norms [2]. South Korea, characterized by a highly competitive and stressful environment, experiences a particularly pronounced deterioration in sleep health among its population [3].

One significant factor compromising sleep quality is sleep apnea, a disorder characterized by recurrent breathing interruptions during sleep. This condition is increasingly prevalent globally, including in South Korea, likely fueled by the adoption of Westernized dietary habits and lifestyles [4]. Sleep apnea is broadly classified into central and obstructive types [5].

Obstructive Sleep Apnea (OSA) is the most common subtype of sleep apnea, which has drawn increasing attention due to its high prevalence. Representing approximately 80–90% of cases, it involves the cessation of breathing due to the physical collapse or obstruction of the upper airway during sleep [6]. According to a systematic review of population-based studies, the prevalence of OSA, defined by an apnea–hypopnea index of ≥5 events per hour, ranges from 9% to 38%, with significantly higher rates observed in men [7].

The insidious nature of OSA, occurring during sleep, often leaves individuals unaware of their condition, highlighting the importance of awareness among family members and healthcare providers to mitigate the risk of secondary complications stemming from airway obstruction [8,9]. In addition to its episodic breathing disruptions, OSA is often accompanied by chronic insufficient sleep duration. According to the American Academy of Sleep Medicine and the Sleep Research Society, adults are recommended to sleep at least 7 h per night to promote optimal health, with shorter durations linked to increased risks of cardiovascular disease, diabetes, and mortality [10].

South Korea presents a notable prevalence of OSA. A nationwide survey revealed that 15.8% of adults are at high risk for OSA, with a higher incidence observed among older individuals and those with elevated body mass index [11].

Despite this, public awareness of sleep apnea syndrome remains limited, underscoring the need for enhanced education and screening initiatives [12]. Obesity has been identified as a major risk factor for OSA, and its prevalence varies according to sociodemographic and socioeconomic characteristics [13]. In response to the growing need for OSA management, continuous positive airway pressure (CPAP) therapy has been introduced and increasingly utilized in South Korea [14]. According to the American Academy of Sleep Medicine (AASM), CPAP remains the first-line treatment for OSA due to its robust efficacy [15]. However, for patients who are intolerant to CPAP, the AASM recommends a personalized approach involving alternative options such as mandibular advancement devices, positional therapy, lifestyle modification, and, in selected cases, surgical interventions like maxillomandibular advancement or upper airway stimulation [16]. These guidelines underscore the importance of individualized, patient-centered care in improving treatment adherence and outcomes.

OSA is strongly associated with an increased risk of developing serious secondary health conditions, including cardiovascular disease and diabetes [17]. Emerging evidence also implicates OSA in adverse oral health outcomes. A comprehensive review of the literature demonstrated significant associations between OSA and a spectrum of oral health issues, such as bruxism, xerostomia (dry mouth), periodontal disease, temporomandibular joint disorders, palatal and dental alterations, and even changes in taste sensation [18,19,20].

Malocclusion, particularly skeletal abnormalities such as mandibular retrognathia, maxillary deficiency, or narrow maxilla, have also been identified as key anatomical risk factors for OSA due to their compromising effect on upper airway space during sleep. In severe cases, surgical approaches such as maxillomandibular advancement (MMA) have been shown to significantly improve airway patency and alleviate OSA symptoms, particularly among patients with craniofacial anomalies. These findings support the growing recognition of structural orofacial contributions to sleep-disordered breathing and highlight the importance of evaluating skeletal morphology in OSA risk assessment [21].

Importantly, recent shifts in the conceptualization of oral health emphasize not only clinical conditions but also functional capacities that directly affect daily life. Rather than being limited to clinical indicators or disease absence, oral health is now recognized to include the ability to perform essential functions, such as chewing and speaking, without pain or discomfort. This broader definition, endorsed by both the World Health Organization and the FDI World Dental Federation, positions oral function as a core component of oral health [22,23]. Within this framework, chewing discomfort, speaking discomfort, and dental pain, which are referred to as oral function problems, are increasingly acknowledged as meaningful indicators of oral health-related quality of life (OHRQoL) [24,25].

Although the relationship between OSA and clinical oral disease has been relatively well documented, its potential impact on oral function remains underexplored. Recent population-based evidence suggests that this link warrants closer attention. A nationally representative Korean study found that individuals at high risk of OSA had significantly higher odds of experiencing dental pain [26]. Additional research has shown that greater OSA severity is associated with orofacial pain, which may interfere with basic oral functions such as chewing and speaking [27]. Collectively, these findings point to a need for a more comprehensive understanding of how OSA may contribute not only to clinical oral pathologies but also to functional impairments that directly affect everyday quality of life.

Despite the recognized significance of oral function in maintaining overall health and quality of life, research directly examining the association between OSA and oral function remains limited. While the effects of OSA on systemic conditions and clinical oral disease have been widely studied [28,29,30], much less attention has been given to its potential impact on the functional aspects of the oral system. Existing studies have instead explored how sleep duration or overall sleep quality relates to outcomes like chewing ability and oral discomfort with a paucity of research specifically addressing the effects of OSA [24,25,31].

Furthermore, several studies have acknowledged the link between OSA and the socioeconomic disadvantage [32,33,34], few have focused specifically on how these disparities manifest in functional oral health outcomes. A systematic review confirmed that individuals with lower income levels are at consistently higher risk of OSA across diverse populations [32], and recent evidence also points to income disparities in OSA prevalence [34,35].

Therefore, research investigating the impact of OSA on oral function among Koreans, as well as within these specific vulnerable populations is particularly lacking. To address this gap, this study analyzes nationally representative data from the Korea National Health and Nutrition Examination Survey (KNHANES) to examine the relationship between OSA and oral function. Specifically, this study investigates the prevalence of OSA and oral function problems among Korean adults aged 40 and older. It also explores how oral function varies according to socioeconomic and health-related factors, and assesses whether individuals with higher OSA risk are more likely to experience problems such as chewing discomfort, speaking difficulty, and dental pain. Furthermore, the study examines whether these associations are more pronounced in low-income or low-education groups to determine if OSA has a disproportionate impact on vulnerable populations.

Grounded in these objectives, we established the following hypotheses: (H1) obstructive sleep apnea (OSA) is significantly associated with oral function problems; (H2) individuals with a higher risk of OSA are more likely to experience greater difficulties in chewing, speaking, and dental pain; and (H3) these associations are more prominent among socioeconomically disadvantaged groups, particularly those with lower education or household income (see Appendix A).

Through this research, we endeavor to identify and explore continuous management strategies for OSA, particularly in vulnerable populations, and to emphasize the risks associated. By elucidating the impact of OSA on oral function, we aim to contribute to the enhancement of individual health and the broader public health landscape, ultimately advocating for policy support to improve health outcomes across society.

## 2. Materials and Methods

### 2.1. Data Source and Study Population

For this study, data were collected from the ninth (2022–2023) Korea National Health and Nutrition Examination Survey (KNHANES). KNHANES is an annual, nationwide population-based cross-sectional survey conducted by the Korea Disease Control and Prevention Agency. It is designed as a complex sample survey using a multistage stratified probability sampling method to ensure national representativeness of the non-institutionalized Korean population. The purpose of the survey is to produce nationally reliable statistics on health status, health behaviors, dietary habits, and nutritional intake, which serve as essential data for developing and evaluating public health policies. KNHANES consists of a health interview, a health examination, and a nutrition survey, making it a validated and comprehensive source for analyzing health-related outcomes among Korean adults.

For this study, we used data from the 9th KNHANES (2022–2023), which initially included 13,194 individuals. As illustrated in Figure 1, 6280 individuals who lacked information on the STOP-Bang score were excluded from analysis. Of the remaining 6914 participants, we excluded an additional 547 individuals due to missing data on oral function problems (chewing discomfort, speaking discomfort, and dental pain). Furthermore, 18 participants with incomplete information on key sociodemographic variables (e.g., education, income, and residence) were excluded. As a result, a final analytic sample of 6349 participants was included in the study.

As this study used secondary data from the 2022–2023 Korea National Health and Nutrition Examination Survey (KNHANES), the sample size was not determined through an a priori power calculation. Instead, the final analytical sample (*n* = 6349) reflects the number of individuals who met the inclusion criteria and had complete data on key study variables. Based on established thresholds for detecting small effect sizes (g = 0.1), a recent meta-research study suggests that approximately 2180 participants are required to achieve 90% statistical power [36]. Therefore, the sample size in the present study is considered sufficient to detect even small associations with high power.

### 2.2. Measurements

The independent variable is obstructive sleep apnea (OSA) measured by the STOP-Bang score. The STOP-Bang score is an 8-item questionnaire which is a widespread tool to evaluate the risk of OSA. STOP-Bang score is an abbreviation for the eight risk factors associated with OSA (see Appendix A). The eight risk factors are snoring, tiredness during daytime, observed apnea, high blood pressure, BMI (≥35 kg/m^2^), age (≥50 years), neck circumference (≥40 cm), and gender (male). This study adjusted BMI and neck circumference to the Asian population according to previous studies [18]. BMI was adjusted from 35 kg/m^2^ to 30 kg/m^2^, according to the severe obesity stage of the Korea Centers for Disease Control and Prevention (KCDC). The neck circumference was also adjusted from 40 cm to 36.3 cm, following the standards for Asians. Each positive response to the questionnaire was given one point and then summed up to a maximum scale of eight points. Based on the measured scores, participants were classified into OSA low risk (0–2 points) and OSA high risk (3–8 points) [37].

The dependent variable, oral function, was measured using self-reported items on chewing discomfort, speaking discomfort, and dental pain. These variables have been validated in prior studies as meaningful indicators of oral function and have been widely used in population-based assessments of oral health and quality of life [24,25]. For chewing discomfort, participants were asked “Currently, do you have chewing discomfort due to problems related in your mouth, such as teeth, dentures, or gums? (If you use dentures, please tell us how you feel while wearing them)”. Participants’ responses were checked on a 5-point Likert scale: “severe discomfort”, “discomfort”, “moderate”, “no discomfort”, and “no discomfort at all”. The conversion of the Likert scale responses into binary categories (yes/no) was based on established cut-off points used in previous population health studies, which identified “severe discomfort” and “discomfort” as clinically meaningful indicators of oral function impairment [23,24,25]. Accordingly, “severe discomfort” and “discomfort” were classified as “yes”, and the remaining responses as “no”. Speaking discomfort was classified using the same approach, based on the same 5-point Likert response scale. To evaluate dental pain, participants were asked “Have you experienced dental pain in the past year? (e.g., aching, throbbing, or dental pain, or pain when consuming cold or hot drinks or food)”. The responses were divided into “yes” or “no”.

Covariates were set as socioeconomic factors and health-related factors. Sex was classified as male and female. Age was classified as 40s, 50s, 60s, and over 70s. Residence was classified as urban and rural. Occupation was classified as white collar, blue collar, and no occupation. Household income was classified from quartile 1 (lowest income) to quartile 4 (highest income). Education level was classified as very low (≤Primary School), low (Middle School), moderate (High School), and high (≥College). Drinking status was classified as yes, or no. Smoking status was classified as current, past, or never. BMI (kg/m^2^) was classified as underweight (<18.4), normal (18.5–22.9), overweight (23.0–24.9), and obesity (≥25). Sleeping duration was classified as very insufficient (<6 h/day), insufficient (6 to <7 h/day), sufficient (7 to <9 h/day), and excessive (≥9 h/day). Hypertension and diabetes were classified as yes or no.

### 2.3. Statistical Analysis

As KNHANES utilizes a structured survey design, data analyses were performed through complex sample analysis, including weights, hierarchical variables, and cluster variables. Frequency analysis was conducted to analyze the distributions, and chi-square test was conducted to examine the relationship between OSA and oral health. Logistic regression analyses were employed to estimate odds ratios (ORs) and 95% confidence intervals (CIs) for the association between OSA and oral function, with ORs interpreted as measures of effect size to assess clinical relevance. To further explore whether the OSA–oral function relationship varied by socioeconomic status, stratified logistic regressions were conducted by education level and household income. All analyses were performed using SPSS version 23.0 (SPSS Inc., Chicago, IL, USA), with statistical significance set at *p* < 0.05.

## 3. Results

### 3.1. General Characteristics of the Participants According to Oral Function Problems

Table 1 presents the general characteristics of the 6349 participants stratified by the presence or absence of oral function problems, specifically chewing discomfort, speaking discomfort, and dental pain. Overall, 1359 participants reported chewing discomfort, 443 reported speaking discomfort, and 1852 reported dental pain. These oral function problems were more frequently reported among females, older adults, rural residents and individuals with lower socioeconomic status.

Educational level and household income showed a clear gradient in the prevalence of oral function problems. Among those with very low education, 35.6% reported chewing discomfort, 14.3% reported speaking discomfort, and 29.6% reported dental pain. In contrast, participants with high education reported much lower rates, with 12.2% for chewing discomfort, 2.2% for speaking discomfort, and 21.4% for dental pain. Similarly, oral function problems were disproportionately concentrated in lower-income groups. In the lowest income quartile (Q1), 36.0% experienced chewing discomfort, 14.8% reported speaking discomfort, and 31.8% experienced dental pain, compared to only 12.6%, 1.7%, and 18.6%, respectively, in the highest income quartile (Q4).

### 3.2. Association Between Obstructive Sleep Apnea (OSA) and Oral Function Problems

Table 2 demonstrates the association between OSA risk and oral function problems. Chewing discomfort was more frequently reported among participants at high risk for OSA (27.7%) compared to those at low risk (18.0%), with a highly significant difference (*x*^2^ = 74.972, *p* < 0.001). Similarly, speaking discomfort was reported by 9.8% of the high-risk group, compared to only 5.5% of the low-risk group (*x*^2^ = 38.972, *p* < 0.001). In addition, the prevalence of dental pain was also higher among those with high OSA risk (31.6%) than among those with low risk (27.8%), and this difference reached statistical significance (*x*^2^ = 10.178, *p* < 0.01). These findings show a statistically significant association between OSA risk and each oral function problem, indicating that individuals at higher risk of OSA are more likely to experience difficulties in chewing, speaking, and dental pain.

### 3.3. Odds Ratios for the Association Between Obstructive Sleep Apnea (OSA) Risk and Oral Function Problems

Table 3 presents the results of multivariable logistic regression analyses examining the association between OSA and three oral function problems (see Appendix A). After adjusting for all covariates, individuals at high risk of OSA had significantly higher odds of reporting chewing discomfort (OR = 1.365, 95% CI = 1.121–1.662), speaking discomfort (OR = 1.534, 95% CI = 1.126–2.082), and dental pain (OR = 1.198, 95% CI = 1.006–1.431), compared to those at low risk. Educational level showed a strong gradient with oral function problems. Participants with very low education had the highest odds of reporting chewing discomfort (OR = 2.078, 95% CI = 1.595–2.707) and speaking discomfort (OR = 3.308, 95% CI = 1.884–4.905), compared to those with high education. A similar trend was observed for moderate and low education levels. Meanwhile, the association between OSA and dental pain showed a different trend across education levels; however, these differences were not statistically significant. Household income was also associated with oral function problems. Participants in the lowest income quartile (Q1) had significantly higher odds of chewing discomfort (OR = 1.942, 95% CI = 1.510–2.496), speaking discomfort (OR = 3.308, 95% CI = 2.066–5.310), and dental pain (OR = 1.477, 95% CI = 1.177–1.855), compared to those in the highest quartile (Q4). The association was weaker but still statistically significant in Q2 and Q3 for some outcomes.

### 3.4. Stratified Analysis Based on Socioeconomic Variables

Figure 2 displays the adjusted odds ratios (ORs) and 95% confidence intervals (CIs) for the association between education level, household income, and oral function problems, namely chewing discomfort, speaking discomfort, and dental pain (see Appendix A).

For chewing discomfort, individuals with very low education had significantly higher odds compared to those with high education (OR = 1.983, 95% CI = 1.357–2.897), as did those in the lowest income quartile (Q1 vs. Q4: OR = 1.996, 95% CI = 1.365–2.918). Education and income gradients were generally observed, though not all comparisons reached statistical significance.

For speaking discomfort, the association was stronger. Very low education (OR = 3.798, 95% CI = 1.759–8.200) and Q1 income (OR = 5.207, 95% CI = 2.365–11.462) were both significantly associated with higher odds of reporting speaking discomfort, with a clear dose–response pattern observed across both education and income levels.

For dental pain, educational level was not significantly associated with the outcomes. However, individuals in the first income quartile (Q1 vs. Q4) showed a statistically significant higher likelihood of experiencing dental pain (OR = 1.486, 95% CI = 1.053–2.098).

These results suggest that individuals with lower socioeconomic status are more likely to experience oral discomfort, particularly in functional aspects such as chewing and speaking.

## 4. Discussion

This study utilized data from the Korea National Health and Nutrition Examination Survey (KNHANES) to analyze the correlation between obstructive sleep apnea (OSA) and oral function. While previous studies have primarily focused on the relationship between OSA and clinical oral health conditions such as periodontal disease or xerostomia, few studies have examined how OSA may affect oral function which includes aspects of daily life such as chewing, speaking, and experiencing dental pain. By expanding the conventional definition of oral health to encompass functional components, this study provides a broader understanding of the impact of OSA on oral well-being.

Firstly, this study utilizes the latest large-scale national database, the 9th Korea National Health and Nutrition Examination Survey (KNHANES), to enhance the generalizability of the results. KNHANES is designed as a complex sample dataset, allowing it to represent the entire Korean population and enhancing the accuracy and validity of the results compared to other datasets, which are small and localized. Furthermore, while previous studies used NHANES and older KNHANES datasets, they did not include the most recent trends or health outcomes in the population [26,38]. This study, with 6349 participants from a nationwide sample, represents the latest data, ensuring current and statistically robust findings that are representative of the Korean adult population.

Secondly, this study assessed OSA risk using the STOP-Bang questionnaire, adjusting the BMI and neck circumference criteria to align with the characteristics of the Asian population, as suggested by Pavarangkul et al. [39]. This adjustment is crucial because applying Western standards to Asian populations can lead to the underdiagnosis of OSA. Additionally, some previous studies relied on subjective, questionnaire-based OSA assessments or simplified sleep recording devices, potentially compromising accuracy. For example, Loke et al. [40] used a questionnaire that relied on subjective judgments, which could lead to lower accuracy in OSA diagnosis. This study, however, utilized a more objective STOP-Bang score, adjusted for the Asian population based on Kang et al. [41], by modifying the body mass index (BMI) and neck circumference criteria to improve the accuracy of OSA assessment and enhance the relevance of findings for the Korean population.

Additionally, this study employed a comprehensive approach to measuring oral health by examining not only the presence of periodontal disease or cavities, but also discomfort in chewing, speaking, and dental pain, capturing a wider range of oral health issues experienced in daily life. While many existing studies have assessed oral health primarily in terms of clinical indicators such as periodontal disease or dental caries [28,42], relatively few have examined how OSA may influence oral function. However, this study aims to capture the multifaceted impact of OSA on oral function by assessing these various aspects. The ability to chew, speak, and live without dental pain constitutes a critical dimension of oral health, particularly in relation to quality of life. Recent international frameworks, including those by the WHO and FDI, emphasize oral function as an essential part of oral health, extending beyond disease-based models [23,24]. Reflecting this perspective, the present study assessed oral function through indicators like chewing, speaking discomfort and dental pain, offering a more holistic understanding of how OSA may be associated with daily oral functioning [24,25].

Lastly, this study focused on socioeconomically disadvantaged groups, especially those with low income or education levels, and conducted stratified analyses. While Movahed et al. [38] briefly considered socioeconomic factors, the analysis was limited to general associations without detailed subgroup examinations. In contrast, in this study stratified analyses were conducted to provide a more nuanced understanding of how social vulnerability exacerbates the relationship between OSA and poor oral function. These findings offer actionable insights into addressing health disparities and crafting targeted public health policies.

In conclusion, this study advances the field by utilizing up-to-date, representative data, refining methodological approaches for the Asian population, and focusing on socio-economically vulnerable groups. These strengths address gaps in prior research and contribute meaningful insights for both academic understanding and public health policy development.

This study reveals that obstructive sleep apnea (OSA) is associated with an increased likelihood of various oral health problems, including difficulties with chewing, speaking, and dental pain. Notably, the relationship of OSA on oral health was more pronounced in low-income and low-education groups. These findings highlight the complex interplay between OSA and oral health, with significant implications for oral health inequalities, particularly among vulnerable populations.

While previous research has explored the relationship between OSA and clinical oral diseases such as periodontal disease or xerostomia [28,42], functional aspects of oral health have received less attention. However, despite establishing a link between periodontal disease and OSA, previous research often overlooked functional oral health issues such as difficulties with chewing or speaking [43]. This study addresses this gap by including these functional aspects, thereby shedding light on the everyday impact of oral problems that prior research may have missed. For instance, we emphasize that problems like dental pain may affect not only physical health but also social and psychological well-being. This finding provides valuable insights into developing more comprehensive public health policies.

Furthermore, although some studies have confirmed the association between OSA and periodontal disease through meta-analysis, they often lacked a thorough examination of socioeconomic factors [44]. This study addresses this limitation by providing a more detailed analysis of the interplay between OSA and oral function within vulnerable populations.

Previous research has shown an increased risk of periodontal disease in OSA patients. Although this study did not directly assess periodontal outcomes, it contributes to this line of evidence by highlighting related oral function problems that may also reflect underlying oral health deterioration in OSA patients [43]. This has also been confirmed in studies by Seo [28] and Kornegay [29], where both found a significantly higher prevalence or risk of periodontal disease in OSA patients. While these studies have contributed valuable evidence on the pathological consequences of OSA, they focused on clinical oral disease. In contrast, the present study expands the perspective by examining functional aspects of oral health that affect daily life, such as chewing, speaking discomfort and dental pain. This approach highlights the broader impact of OSA on quality of life, extending beyond traditional disease-based outcomes. This is further supported by previous studies that have reported associations between OSA and poorer oral health-related quality of life (OHRQoL) and xerostomia, underscoring the broader impact of OSA on oral well-being [30,45].

It has been reported that oral function is generally worse in socioeconomically disadvantaged groups [24,25,30,46]. This study reinforces this finding by highlighting the impact of OSA on these vulnerable populations, further strengthening the credibility of our results. Notably, the stronger associations between OSA and oral function observed in low-income and low-education groups suggest the need for more targeted public health interventions. This is consistent with the findings of Etindele [32] and Redline [33], which showed that lower socioeconomic status is associated with a higher prevalence of OSA and lower access to treatment, and that the prevalence and severity of sleep-disordered breathing vary by race and ethnicity. While some studies have reported an association between OSA and oral pain, they failed to establish a significant link with chewing discomfort [26]. In contrast, this study indicates a significant relationship between OSA and all aspects of chewing, speaking, and dental pain, thus complementing previous research and enhancing the validity of our findings.

The findings of this study offer several important implications for policy and public health. Firstly, it highlights the need to apply appropriate criteria for OSA screening in Asian populations. Using existing Western criteria without adjustments may reduce diagnostic sensitivity for Asians. Therefore, screening tools like the STOP-Bang questionnaire should be modified to reflect the specific characteristics of this population, thereby improving the effectiveness of early diagnosis and prevention.

Secondly, public health campaigns emphasizing the combined risk of OSA and oral function problems in low-income and low-education groups can inform the design of early prevention and management programs. Low-income individuals often face economic burdens, lack of medical information, and limited health awareness, reducing their likelihood of initiating OSA treatment like continuous positive airway pressure (CPAP) [47]. Similarly, parents with lower education levels may have limited awareness of their children’s sleep apnea, making it difficult to provide appropriate care [34]. These campaigns can be community-based and incorporate comprehensive health checkups that include oral health screenings and OSA risk assessments. Furthermore, since vulnerable populations are more likely to struggle with creating a healthy sleep environment, education and support are needed for them. It is crucial to establish systems within public healthcare institutions and local health centers to effectively screen individuals at high risk for OSA and provide appropriate treatment.

Moreover, long-term observational studies and public health policy development that link OSA and oral function problems should be pursued through national-level data collection and analysis. This approach can lead to improvements in the overall health of communities and the nation while reducing health disparities among vulnerable populations.

In addition, given the limitations of self-reported screening tools, future public health strategies should consider integrating polysomnography into diagnostic protocols for high-risk individuals. As a gold-standard method, polysomnography provides comprehensive and objective data on brain activity, oxygen saturation, breathing patterns, and muscle movement during sleep, allowing for more accurate diagnosis of OSA [8]. Implementing polysomnography-based screening, especially within public health institutions, can facilitate early and precise identification of OSA among vulnerable populations, thereby improving the targeting of treatment resources and reducing the burden of undiagnosed cases. 

This study has several limitations. Firstly, this study is a cross-sectional study, which limits examining the causal relationship between OSA and oral health. It is difficult to analyze and understand changes over time regarding OSA and its impact on oral health because the data were collected at a single point in time. As a result, while this study identifies significant correlations, caution should be exercised when interpreting these findings. Future research needs to address this limitation by utilizing a longitudinal design.

Secondly, this study’s subjects were limited to individuals aged 40 and older, excluding younger age groups from the study. Since OSA and related oral health problems can also occur in those under 40, future research should include a broader age range to provide a more comprehensive understanding.

Thirdly, data were collected through self-reported questionnaires, which may introduce recall bias. In the case of OSA symptoms, relying on individuals’ awareness or memory may lead to inaccuracies in the data. In addition, responses may vary depending on individual perception, cognitive awareness, or pain sensitivity, potentially introducing information bias. Such biases could affect the accuracy of OSA diagnosis and oral function problems. Although the current study used the STOP-Bang questionnaire, future research could benefit from using polysomnography, a more objective diagnostic tool, to enhance the accuracy of OSA assessment and strengthen causal inference [8].

Fourthly, this study assesses oral function using chewing discomfort, speaking discomfort, and dental pain, which limited the diversity of oral health variables. Important aspects of oral health, such as periodontal disease and dental caries, were not included, hindering a comprehensive evaluation of the relationship between OSA and oral health. This limitation was partly due to the characteristics of the dataset, as the KNHANES oral health data did not include clinical assessments of periodontal status or caries. Nevertheless, although the relationship between the prevalence of actual oral diseases such as periodontitis could not be determined, this study provides a broader view of oral health by focusing on functional discomforts, compared to previous studies that examined only a narrow set of clinical indicators.

Additionally, although major covariates such as age, sex, smoking, alcohol consumption, and chronic diseases were adjusted for, other confounding variables may not have been fully controlled. These include oral hygiene behaviors, education level, access to dental care, as well as nutrient intake such as vitamins and calcium, which can influence both OSA and oral function. Future studies should include these variables to provide a more comprehensive understanding of the relationship and to strengthen the validity of the findings.

Finally, the study was conducted on Korean population, which limits the generalizability of the findings to other countries and ethnicities. Therefore, further research, including diverse countries and ethnicities, is needed.

Despite these limitations, this study offers several important strengths. By utilizing data from the KNHANES, it provides valuable insights specific to the Korean population and addresses a gap in the literature, which has been largely based on Western datasets and criteria. The analysis reflects physical and demographic characteristics more relevant to Asian populations, enhancing the contextual validity of the findings. Moreover, this study broadens the understanding of oral health by examining functional aspects such as chewing discomfort, speaking difficulty, and dental pain, rather than focusing solely on clinical diagnoses. This approach presents a more comprehensive perspective on the impact of OSA on daily oral function.

Another strength of the study lies in its emphasis on health equity. By analyzing differences across income and education levels, it highlights the disproportionate burden of OSA among socioeconomically disadvantaged groups and reinforces the need for targeted screening and management strategies. These findings contribute to the field of public health by linking clinical symptoms to broader systemic inequalities and offering evidence to support early detection and the development of equitable healthcare policies.

By these points, this study underscores the importance of enhancing accessibility to OSA screening and management for vulnerable populations. By notifying the difficulties faced by socially disadvantaged groups, it highlights the importance of early detection and targeted intervention strategies to reduce health disparities and promote equitable healthcare systems. Our findings provide valuable insights for public health policymakers to address health inequality and develop equitable healthcare strategies. Eventually, this study fills the gap between clinical evidence and public health policy, offering meaningful contributions to advancing public health and promoting health equity.

## 5. Conclusions

This study investigated the association between obstructive sleep apnea (OSA) and oral function, specifically discomfort in chewing, speaking, and dental pain, using data from the Korea National Health and Nutrition Examination Survey (KNHANES). The findings revealed a significant association between OSA and poorer oral function outcomes. Individuals at high risk for OSA had greater odds of experiencing chewing discomfort (OR = 1.365, 95% CI = 1.121–1.662), speaking discomfort (OR = 1.534, 95% CI = 1.126–2.082), and dental pain (OR = 1.198, 95% CI = 1.006–1.431). These associations were particularly pronounced among the most socioeconomically disadvantaged groups: participants with very low education had almost twice the odds of chewing discomfort (OR = 1.983, 95% CI = 1.357–2.897), and those in the lowest income quartile (Q1) had over five times the odds of speaking discomfort (OR = 5.207, 95% CI = 2.365–11.462) compared to their respective reference groups.

These findings highlight the complex interplay between OSA and oral function, particularly within socioeconomically disadvantaged groups, and underscore the importance of integrated health strategies that consider both sleep-related disorders and oral health disparities. Based on this, public health interventions should simultaneously address OSA and oral function, especially in populations with limited access to care. Community-based initiatives that combine sleep assessments with dental checkups and health education may serve as effective tools for early detection and management. Furthermore, multidisciplinary collaboration across sleep medicine, dentistry, and public health policy can help reduce the disproportionate burden of OSA-related oral dysfunction. Such integrated approaches have the potential to not only improve individual well-being but also mitigate structural health inequalities and promote more equitable health outcomes at the population level.

Future research should build upon these findings by employing longitudinal study designs to examine the causal relationship between OSA and oral function over time. Additionally, incorporating objective OSA diagnostic tools, such as polysomnography, could enhance the accuracy of OSA diagnosis and strengthen the validity of the findings. Finally, expanding the range of oral function or health variables measured would contribute to a more holistic understanding of the complex interplay between OSA and oral well-being.

## Figures and Tables

**Figure 1 healthcare-13-01323-f001:**
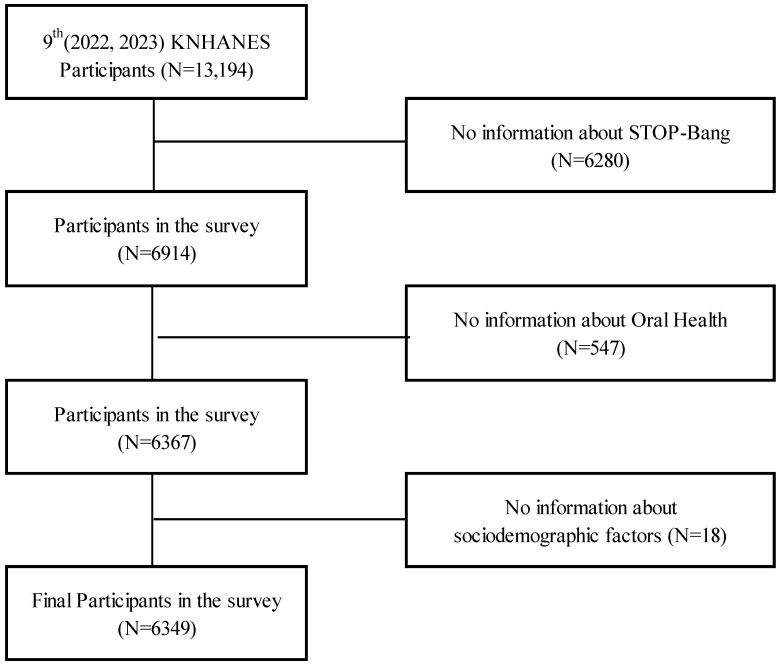
Flowchart of research subject selection.

**Figure 2 healthcare-13-01323-f002:**
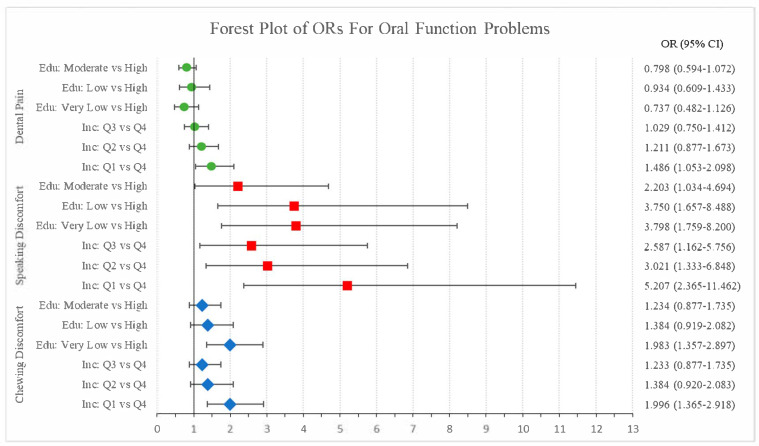
Forest plot of odds ratios for oral function problems. Note: OR = odds ratio; CI = confidence interval. All variables were adjusted for relevant covariates in the regression model. Education level (“Edu”) and household income (“Inc”) are presented as key stratifying variables, with the highest education level (High) and the highest income quartile (Q4) set as reference categories.

**Table 1 healthcare-13-01323-t001:** General characteristics of the participants according to oral function problems.

Characteristics		Chewing Discomfort		Speaking Discomfort		Dental Pain	
	Total (%)	Yes (%)	No (%)	Yes (%)	No (%)	Yes (%)	No (%)
	(*n* = 6349)	(*n* = 1359)	(*n* = 4990)	(*n* = 443)	(*n* = 5906)	(*n* = 1852)	(*n* = 4497)
Gender							
Male	2522 (39.7)	622 (9.8)	1900 (29.9)	214 (3.4)	2308 (36.3)	774 (12.2)	1748 (27.5)
Female	3827 (60.3)	737 (11.6)	3090 (48.7)	229 (3.6)	3598 (56.7)	1078 (17.0)	2749 (43.3)
Age							
40s	1345 (21.2)	142 (2.2)	1203 (19.0)	21 (0.3)	1324 (20.9)	401 (6.3)	944 (14.9)
50s	1528 (24.1)	243 (3.8)	1285 (20.2)	69 (1.1)	1459 (23.0)	456 (7.2)	1072 (16.9)
60s	1895 (29.8)	449 (7.1)	1446 (22.8)	137 (2.2)	1758 (27.7)	569 (9.0)	1326 (20.9)
≥70s	1581 (24.9)	525 (8.3)	1056 (16.6)	216 (3.4)	1365 (21.5)	426 (6.7)	1155 (18.2)
Residence							
Rural	4838 (76.2)	958 (15.1)	3880 (61.1)	297 (4.7)	4541 (71.5)	1377 (21.7)	3461 (54.5)
Urban	1511 (23.8)	401 (6.3)	1110 (17.5)	146 (2.3)	1365 (21.5)	475 (7.5)	1036 (16.3)
Occupation							
White Color	1256 (19.8)	146 (2.3)	1110 (17.5)	25 (0.4)	1231 (19.4)	330 (5.2)	926 (14.6)
Blue Color	2471 (38.9)	553 (8.7)	1918 (30.2)	174 (2.7)	2297 (36.2)	759 (12.0)	1712 (27.0)
No Occupation	2622 (41.3)	660 (10.4)	1962 (30.9)	244 (3.8)	2378 (37.4)	763 (12.0)	1859 (29.3)
Education Level							
very low	1422 (22.4)	507 (8.0)	915 (14.4)	203 (3.2)	1219 (19.2)	421 (6.6)	1001 (15.8)
low	765 (12.0)	209 (3.3)	556 (8.7)	78 (1.2)	687 (10.8)	241 (3.8)	524 (8.2)
moderate	2071 (32.6)	387 (6.1)	1684 (26.5)	115 (1.8)	1956 (30.8)	606 (9.5)	1465 (23.1)
high	2091 (32.9)	256 (4.0)	1835 (28.9)	47 (0.7)	2044 (32.2)	584 (9.2)	1507 (23.7)
House Income							
Q1	1477 (23.2)	531 (8.4)	946 (14.9)	218 (3.4)	1259 (19.8)	470 (7.4)	1007 (15.9)
Q2	1599 (25.2)	357 (5.6)	1242 (19.6)	126 (2.0)	1473 (23.2)	477 (7.5)	1122 (17.7)
Q3	1575 (24.8)	257 (4.0)	1318 (20.8)	70 (1.1)	1505 (23.7)	461 (7.3)	1114 (17.6)
Q4	1698 (26.8)	214 (3.4)	1484 (23.4)	29 (0.5)	1669 (26.3)	444 (7.0)	1254 (19.8)
Drinking							
No	2207 (34.7)	560 (8.8)	1647 (25.9)	205 (3.2)	2002 (31.5)	630 (9.9)	1577 (24.8)
Yes	4142 (65.3)	799 (12.6)	3343 (52.7)	238 (3.8)	3904 (61.5)	1222 (19.2)	2920 (46.0)
Smoking							
Never	3986 (62.8)	744 (11.7)	3242 (51.1)	223 (3.5)	3761 (59.3)	1081 (17.0)	2905 (45.8)
Past	1503 (23.7)	362 (5.7)	1141 (18.0)	121 (1.9)	1382 (21.8)	469 (7.4)	1034 (16.3)
Current	860 (13.5)	253 (4.0)	607 (9.5)	99 (1.6)	760 (12.0)	302 (4.8)	558 (8.8)
BMI ^1^ (kg/m^2^)							
Underweight (<18.4)	221 (3.5)	49 (0.8)	172 (2.7)	22 (0.3)	199 (3.1)	65 (1.0)	156 (2.5)
Normal (18.5–22.9)	2466 (38.8)	517 (8.1)	1949 (30.7)	173 (2.7)	2293 (36.1)	724 (11.4)	1742 (27.4)
Overweight (23.0–24.9)	1643 (25.9)	338 (5.3)	1305 (20.6)	104 (1.6)	1539 (24.3)	447 (7.0)	1196 (18.8)
Obesity (≥25)	2019 (31.8)	455 (7.2)	1564 (24.6)	144 (2.3)	1875 (29.5)	616 (9.7)	1403 (22.1)
Sleeping Duration							
≤6 h	1168 (18.4)	330 (5.2)	838 (13.2)	120 (1.9)	1048 (16.5)	371 (5.8)	797 (12.6)
6–7 h	1686 (26.6)	350 (5.5)	1336 (21.1)	101 (1.6)	1585 (25.0)	502 (7.9)	1184 (18.7)
7–9 h	3200 (50.4)	606 (9.5)	2594 (40.8)	187 (2.9)	3013 (47.4)	893 (14.1)	2307 (36.3)
≥9 h	295 (4.6)	73 (1.2)	222 (3.5)	35 (0.6)	260 (4.1)	86 (1.4)	209 (3.3)
HTN ^2^							
Yes	2158 (34.0)	565 (8.9)	1593 (25.1)	194 (3.1)	1964 (30.9)	623 (9.8)	1535 (24.2)
No	4191 (66.0)	794 (12.5)	3397 (53.5)	249 (3.9)	3942 (62.1)	1229 (19.4)	2962 (46.7)
DM ^3^							
Yes	886 (13.9)	254 (4.0)	632 (9.9)	87 (1.4)	799 (12.6)	270 (4.3)	616 (9.7)
No	5463 (86.1)	1105 (17.4)	4358 (68.6)	356 (5.6)	5107 (80.4)	1582 (24.9)	3881 (61.1)

Note: BMI ^1^: Body Mass Index; HTN ^2^: Hypertension; DM ^3^: Diabetes Mellitus.

**Table 2 healthcare-13-01323-t002:** Association between obstructive sleep apnea (OSA) and oral function problems.

Variables		Chewing Discomfort			Speaking Discomfort			Dental Pain		
	Total (%)	Yes (%)	No (%)	*x* ^2^	Yes (%)	No (%)	*x* ^2^	Yes (%)	No (%)	*x* ^2^
	(*n* = 6349)	(*n* = 1359)	(*n* = 4990)		(*n* = 443)	(*n* = 5906)		(*n* = 1852)	(*n* = 4497)	
OSA ^1^										
Low Risk	4126 (65.0)	743 (11.7)	3383 (53.3)	74.972 ***	226 (3.6)	3900 (61.4)	38.972 ***	1149 (18.1)	2977 (46.9)	10.178 **
High Risk	2223 (35.0)	616 (9.7)	1607 (25.3)		217 (3.4)	2006 (31.6)		703 (11.1)	1520 (23.9)	

Note: ** *p* < 0.01 *** *p* < 0.001; OSA ^1^: Obstructive Sleep Apnea. All models were adjusted for gender, age, residential area, occupation, education level, household income, drinking status, smoking status, body mass index (BMI), sleep duration, hypertension, and diabetes.

**Table 3 healthcare-13-01323-t003:** Results of logistic regression analysis of the association between obstructive sleep apnea and oral function problems.

Variables	Oral Health		
	Chewing Discomfort	Speaking Discomfort	Dental Pain
	OR (95% CI^2^)	OR (95% CI)	OR (95% CI)
OSA ^1^			
Low Risk	1	1	1
High Risk	1.365 (1.121–1.662) **	1.534 (1.126–2.082) **	1.198 (1.006–1.431) *
Gender			
Male	1	1	1
Female	1.025 (0.813–1.292)	1.077 (0.744–1.554)	1.226 (0.988–1.514)
Age			
40s	1	1	1
50s	1.290 (1.006–1.657) *	1.862 (1.086–3.247) ***	0.988 (0.810–1.202)
60s	1.673 (1.274–2.197) ***	1.876 (1.094–3.177) ***	0.885 (0.714–1.095)
≥70s	2.016 (1.470–2.766) ***	2.815 (1.535–5.202) **	0.673 (0.494–0.901) **
Residence			
Rural	1	1	1
Urban	1.097 (0.900–1.336)	1.172 (0.883–1.556)	1.116 (0.866–1.437)
Occupation			
White Color	1	1	1
Blue Color	1.116 (0.848–1.470)	1.216 (0.779–2.224)	1.248 (1.031–1.509) *
No Occupation	1.036 (0.789–1.360)	1.314 (0.716–2.061)	1.128 (0.927–1.373)
Education Level			
Very Low	2.078 (1.595–2.707) ***	3.308 (1.884–4.905) ***	0.878 (0.682–1.129)
Low	1.559 (1.178–2.062) **	2.785 (1.705–4.556) ***	1.009 (0.776–1.312)
Moderate	1.261 (1.028–1.547) *	2.033 (1.320–3.133) **	0.905 (0.756–1.083)
High	1	1	1
House Income			
Q1	1.942 (1.510–2.496) ***	3.308 (2.066–5.310) ***	1.477 (1.177–1.855) **
Q2	1.321 (1.053–1.657) *	2.211 (1.382–3.533) **	1.169 (0.965–1.417)
Q3	1.104 (0.880–1.386)	1.668 (1.046–2.656) *	1.155 (0.967–1.379)
Q4	1	1	1
Drinking			
No	0.907 (0.754–1.092)	0.844 (0.654–1.089)	1.014 (0.893–1.151)
Yes	1	1	1
Smoking			
Never	1	1	1
Past	1.489 (1.173–1.890) **	1.680 (1.154–2.433) **	1.561 (1.277–1.909) ***
Current	2.227 (1.720–2.882) ***	2.903 (1.957–4.308) ***	1.507 (1.194–1.901) **
BMI ^3^ (kg/m^2^)			
Underweight (<18.4)	0.819 (0.549–1.222)	1.300 (0.741–2.272)	1.000 (0.679–1.471)
Normal (18.5–22.9)	1	1	1
Overweight (23.0–24.9)	0.866 (0.717–1.047)	0.906 (0.660–1.243)	0.914 (0.774–1.080)
Obesity (≥25)	1.002 (0.832–1.205)	1.003 (0.735–1.376)	1.042 (0.891–1.271)
Sleeping Duration			
≤6 h	1.282 (1.067–1.540) **	1.271 (0.978–1.637)	1.085 (0.909–1.294)
6–7 h	1.178 (0.988–1.405)	1.008 (0.763–1.321)	1.104 (0.953–1.280)
7–9 h	1	1	1
≥9 h	0.955 (0.686–1.331)	1.153 (0.731–1.799)	0.863 (0.618–1.206)
HTN ^4^			
Yes	0.741 (0.619–0.888) **	0.606 (0.455–0.807) **	0.898 (0.756–1.067)
No	1	1	1
DM ^5^			
Yes	1.173 (0.957–1.438)	1.107 (0.814–1.505)	1.059 (0.871–1.289)
No	1	1	1

Note: * *p* < 0.05 ** *p* < 0.01 *** *p* < 0.001; ^1^ Obstructive Sleep Apnea; ^2^ Confidence Interval; ^3^ Body Mass Index; ^4^ Hypertension; ^5^ Diabetes Mellitus.

## Data Availability

The datasets used and analyzed during the current study are available from the corresponding author upon reasonable request.

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
