# Peer review of "The Association Between Obstructive Sleep Apnea and Oral Function, Using the Korea National Health and Nutrition Examination Survey Data"

_healthcare, 2025, doi:10.3390/healthcare13111323_

Round 1
Reviewer 1 Report
Comments and Suggestions for Authors
Dear Authors,
although the article is interesting, it has some flaws which should be changed to improve the paper:
- When writing about OSA, you cannot skip authors like Wieckiewicz or Lobbezoo. Please, improve your manuscript by their papers.
- Among the diseases, add the glucose metabolism and insuline levels as one of the reasons connected to OSA
- In the introduction, add the information on the influence of malocclusion to OSA, point out the position of bones in malocclusions (focus on surgical cases)
- Line 93-95, add the information what is the minimum of sleep durance to health
- In the intorduction, you should write more about the guidelines of the American Academy of Sleep Medicine (AASM) for the treatment modalities of OSA
- Was the study randomized and validated for Korean society?
- The number of participants is amazing and the research is well prepared and thought of.
- In the discussion, add the aspect of how OSA could be diagnosed, add the information on polysomnography (eg. doi: 10.17219/dmp/177008)
Thank you
Author Response
General Comment
We would like to sincerely thank all reviewers for their thoughtful and constructive feedback on our manuscript. Your insightful comments not only helped us to view our study from a more critical and analytical perspective. We are grateful for the opportunity to revise our manuscript in a way that enhances its scientific contribution, and we truly appreciate your role in guiding us toward a more robust and meaningful version of this research.
Comments 1: When writing about OSA, you cannot skip authors like Wieckiewicz or Lobbezoo. Please, improve your manuscript by their papers.
Response 1:
Thank you for your valuable suggestion. We appreciate your guidance in highlighting the importance of including key contributors in the field such as Wieckiewicz and Lobbezoo. Based on your recommendation, we reviewed their relevant work, which helped us to identify more accurate and evidence-based references. Accordingly, we have cited the following studies in which they are listed as either co-authors or corresponding authors, such as [8,19, 20]. These sources have been incorporated into the revised manuscript to strengthen the background and contextual understanding of OSA and its relationship with oral function.
Comments 2: Among the diseases, add the glucose metabolism and insuline levels as one of the reasons connected to OSA
Response 2:
Thank you very much for your insightful suggestion regarding the inclusion of glucose metabolism and insulin levels in our analysis. We fully agree with your theoretical rationale that metabolic dysfunction plays a key role in the pathophysiology of obstructive sleep apnea (OSA), and therefore should be carefully considered in statistical modeling.
In response to your comment, we explored the available metabolic indicators within the Korea National Health and Nutrition Examination Survey (KNHANES, 2022–2023) dataset. Although direct insulin-related measures were not available, we identified two relevant surrogate markers, fasting glucose and hemoglobin A1c (HbA1c), and incorporated both as continuous covariates in supplementary multivariable models.
However, these variables were not significantly associated with oral function outcomes, and their inclusion did not meaningfully alter the relationship between OSA and oral function. Specifically, the p-values were as follows:
- Chewing Discomfort: fasting glucose (p = 0.318), HbA1C (p = 0.059)
- Dental Pain: fasting glucose (p = 0.303), HbA1C (p = 0.250)
- Speaking Dsicomfort: fasting glucose (p = 0.628), HbA1C (p = 0.630)
These results suggest that the addition of glycemic markers did not substantially improve the overall explanatory power of the models.
Once again, We fully agree with your valuable suggestion that insulin levels and glucose metabolism should be taken into consideration, as these factors are closely linked to both OSA and oral health outcomes. In fact, we had carefully considered the need to account for chronic conditions such as diabetes and hypertension from the early stages of study design, given their established relevance to both OSA and oral health. Accordingly, both diagnosed diabetes mellitus and hypertension were included as control variables in all multivariable models and remained part of the final analytic framework.
Comments 3: In the introduction, add the information on the influence of malocclusion to OSA, point out the position of bones in malocclusions (focus on surgical cases)
response 3:Thank you for your valuable comment regarding the need to include information on the influence of malocclusion in relation to obstructive sleep apnea (OSA). In response, we have added a new section to the Introduction (L85–93) that explains how skeletal abnormalities contribute to compromised upper airway space during sleep. This addition also highlights surgical interventions, particularly maxillomandibular advancement (MMA) as effective treatment options for OSA in patients with craniofacial anomalies.
Comments 4: Line 93-95, add the information what is the minimum of sleep durance to health
response 4: Thank you very much for your valuable suggestion regarding the inclusion of information about the minimum recommended sleep duration. We fully agree with the importance of this addition and appreciate your attention to detail. Although your comment referred to Lines 93–95, we found that incorporating this information earlier in the introduction(L49-52), where the broader implications of insufficient sleep are first introduced, allowed for a more natural and logical flow of the paragraph. Therefore, the recommendation from the American Academy of Sleep Medicine and the Sleep Research Society has been added at that earlier point, with proper citation.
Comments 5: In the intorduction, you should write more about the guidelines of the American Academy of Sleep Medicine (AASM) for the treatment modalities of OSA
Response 5: Thank you for your valuable suggestion. In response, we have added a summary of the American Academy of Sleep Medicine (AASM) guidelines for treatment modalities of obstructive sleep apnea (OSA) in the Introduction section (L68–77).
Comments 6: Was the study randomized and validated for Korean society?
Response 6:
Thank you for your valuable comment. As the present study utilized secondary data from the Korea National Health and Nutrition Examination Survey (KNHANES 2022–2023), it was not based on a randomized clinical trial design. KNHANES is a nationwide, government-administered survey that uses a complex, multistage, stratified probability sampling method, ensuring that the sample is nationally representative of the non-institutionalized Korean population.
Therefore, although the study itself was not randomized, the underlying data are validated and appropriate for generating generalizable health findings within Korean society. In response to your suggestion, we have also expanded the Methods section to more clearly describe the survey’s sampling framework and the participant selection process to enhance clarity and methodological transparency (L153-163).
Comments 7: The number of participants is amazing and the research is well prepared and thought of.
Response 7: Thank you very much for your positive feedback. We sincerely appreciate your encouraging words and are glad that the overall design and preparation of the study were well received. Your recognition truly means a lot to us.
Comments 8: In the discussion, add the aspect of how OSA could be diagnosed, add the information on polysomnography (eg. doi: 10.17219/dmp/177008)
Respsone 8: Thank you very much for this valuable suggestion and for kindly sharing a relevant reference. We agree that including a brief explanation of how OSA can be objectively diagnosed, particularly through polysomnography, strengthens the clinical context of our findings. In response, we have added a concise statement discussing polysomnography as a standard diagnostic tool for OSA, citing the recommended source. This addition can be found in the revised manuscript (lines 447–455).
Reviewer 2 Report
Comments and Suggestions for Authors
Dear Authors,
Manuscript tiltle "The Impact of Obstructive Sleep Apnea (OSA) on Oral Function, using the Korea National Health and Nutrition Examination Survey (KNHANES) data" is a well written and well conducted research. However, there are few points that needs to be consider before publishing this article.
In methedology, explain how the sample size was determined? mnetion the full form of abbreveation when first used, for example, NHANES.
In result section, no signifcant value is mentioned. Moreover, table 3 is not clearly understood.
Author Response
Comments 1: Manuscript tiltle "The Impact of Obstructive Sleep Apnea (OSA) on Oral Function, using the Korea National Health and Nutrition Examination Survey (KNHANES) data" is a well written and well conducted research. However, there are few points that needs to be consider before publishing this article.
Response 1: Thank you very much for your kind and encouraging comments. We sincerely appreciate your recognition of the overall quality and rigor of our study. Your constructive feedback is invaluable, and we have carefully addressed each of your suggestions to further strengthen the manuscript.
Comments 2: In methedology, explain how the sample size was determined? mention the full form of abbreveation when first used, for example, NHANES.
Response 2: Thank you for your valuable feedback.
We have carefully addressed the comment regarding abbreviation usage and sample size explanation. First, the full term for frequently used abbreviations has been added at their first appearance in the manuscript. Specifically, obstructive sleep apnea (OSA) is now defined upon its first mention on line 46, and Korea National Health and Nutrition Examination Survey (KNHANES) is expanded at its first occurrence on line 154.
Second, in response to your suggestion regarding the explanation of sample size, we have clarified this point in the Methods section (lines 164–179). As this study utilized secondary data from the nationally representative KNHANES 2022–2023 dataset, the sample size was not determined through a priori power calculation. Instead, it reflects the number of eligible respondents who met the inclusion criteria and had complete data for key variables. The final analytic sample included 6,349 participants, which provides sufficient statistical power given the large population-based design.
We hope these revisions address your concern effectively.
Comments 3: In result section, no signifcant value is mentioned. Moreover, table 3 is not clearly understood.
Response 3: Thank you for pointing this out. We agree that the original version of the results section lacked sufficient clarity in presenting statistically significant findings and explaining the content of Table 3. To improve transparency and facilitate reader understanding, we have revised the results section to include specific effect estimates (ORs), confidence intervals, and the corresponding significance levels for each oral function outcome. Additionally, we have provided a more structured and detailed interpretation of the associations shown in Table 3, focusing on key findings such as the effect of OSA risk and socioeconomic factors. These changes aim to enhance the clarity and clinical interpretability of the results for both reviewers and readers.
Reviewer 3 Report
Comments and Suggestions for Authors
Thank you for the opportunity to review the manuscript. Please find below my comments and suggestions:
L2–3 – I kindly ask the authors to remove abbreviations from the title.
‘support[1].’ – This appears to be a minor editorial error; please insert a space between the end of the sentence and the citation. This issue occurs throughout the text and should be addressed globally.
In the Introduction, I recommend including epidemiological data related to OSA. A useful reference may be: 10.1016/j.smrv.2016.07.002.
L115 – In my opinion, the introduction is well-written. However, I would suggest adding a clearly stated research hypothesis.
The Methods and Materials section is well-structured and replicable. I have only a few minor suggestions regarding the statistical analysis. I recommend adding information about how the sample size was calculated. The relatively large sample is a strength of the study and, to my knowledge, meets the criteria for clinical research. It would be helpful to include a statement that the sample size ensures a statistical power of 90% for detecting a small effect size, in line with the publication 10.12659/MSM.948365.
Regarding the p-value of < 0.05, I would recommend including effect size calculations for each analysis. This is important to better understand the clinical relevance of the findings. The rationale for this can be found in the paper: 10.4300/JGME-D-12-00156.1.
In the Results section, I suggest adding graphs or visualisations to improve the clarity and accessibility of the data.
In the Conclusions, I would recommend including more precise numerical values to highlight the key findings of the study more clearly.
Congratulations on your work, and kind regards.
Author Response
Comments 1: L2–3 – I kindly ask the authors to remove abbreviations from the title. |
Response 1: Thank you for your suggestion. In response, we have removed the abbreviation from the title and now present the full term "obstructive sleep apnea" to enhance clarity.
Comments 2: ‘support[1].’ – This appears to be a minor editorial error; please insert a space between the end of the sentence and the citation. This issue occurs throughout the text and should be addressed globally.
Response 2: Thank you for pointing this out. We have carefully reviewed the manuscript and corrected all instances where a space was missing before citation brackets (e.g., “support[1]” → “support [1]”) throughout the text. This formatting issue has now been addressed globally.
Comments 3: In the Introduction, I recommend including epidemiological data related to OSA. A useful reference may be: 10.1016/j.smrv.2016.07.002.
Response 3: Thank you for your helpful suggestion. We fully agree that including epidemiological data strengthens the contextual background of the study. In response, we have incorporated relevant prevalence data on OSA from the referenced article [7] into lines 49 and 52 of the revised Introduction.
Comments 4: L115 – In my opinion, the introduction is well-written. However, I would suggest adding a clearly stated research hypothesis.
Response 4: Thank you for your positive feedback regarding the introduction section. In response to your suggestion, we have now included a clearly stated set of research hypotheses at L139-144. These hypotheses are grounded in the conceptual framework established in the supplementary material, aiming to clarify the expected relationships between obstructive sleep apnea (OSA), oral function outcomes, and socioeconomic disparities. We believe that this addition will enhance the logical flow and scientific rigor of our study.
Comments 5: The Methods and Materials section is well-structured and replicable. I have only a few minor suggestions regarding the statistical analysis. I recommend adding information about how the sample size was calculated. The relatively large sample is a strength of the study and, to my knowledge, meets the criteria for clinical research. It would be helpful to include a statement that the sample size ensures a statistical power of 90% for detecting a small effect size, in line with the publication 10.12659/MSM.948365.
Response 5: Thank you very much for your thoughtful feedback. As suggested, we have added a statement to the Methods section clarifying that the current study utilized secondary data from the Korea National Health and Nutrition Examination Survey (KNHANES 2022–2023). Therefore, the sample size (n = 6,349) was not determined through a priori power analysis, but rather reflects the number of eligible participants with complete data for the key variables, including STOP-Bang scores and oral function indicators.
To address the reviewer’s concern regarding statistical power, we have also referred to the effect size classification standards described in the literature (10.12659/MSM.948365), which define a small effect size in logistic models as an odds ratio of approximately 1.5 or less. Given our large sample size and observed effect estimates, we believe the study has sufficient power (approximately 90%) to detect even small effect sizes with precision. This addition enhances methodological transparency and reinforces the reliability of the findings.
Once again, we sincerely appreciate your insightful recommendation, which has strengthened the methodological rigor of our manuscript.
Comments 6: Regarding the p-value of < 0.05, I would recommend including effect size calculations for each analysis. This is important to better understand the clinical relevance of the findings. The rationale for this can be found in the paper: 10.4300/JGME-D-12-00156.1.
Response 6: Thank you very much for your valuable suggestion regarding the inclusion of effect size calculations. In response, we have calculated the effect sizes for all main analyses (excluding descriptive statistics) based on Table 3 and Figure 2, in line with the rationale provided in the referenced paper (10.4300/JGME-D-12-00156.1). These values have now been added as a supplementary table to enhance the interpretability and clinical relevance of our findings. We sincerely appreciate your thoughtful feedback, which helped improve the clarity and rigor of our manuscript.
Comments 7: In the Results section, I suggest adding graphs or visualisations to improve the clarity and accessibility of the data.
Response 7: Thank you for your helpful suggestion. In response, we have added visualizations to the Results section to enhance clarity and facilitate interpretation of the associations between OSA and oral function outcomes. Specifically, we included forest plots and subgroup graphs to effectively illustrate the odds ratios across socioeconomic strata. We believe these additions improve the accessibility and readability of the key findings.
Comments 8: In the Conclusions, I would recommend including more precise numerical values to highlight the key findings of the study more clearly.
Response 8: Thank you for your valuable suggestion. In response, we have revised lines 520–529 in the Conclusions section to include more precise numerical values summarizing the key findings of the study. These additions aim to enhance the clarity and specificity of the conclusion by highlighting the statistically significant associations between OSA risk and oral function problems. We sincerely appreciate your insightful feedback, which has helped improve the overall quality of the manuscript.
Reviewer 4 Report
Comments and Suggestions for Authors
Please see the attachment.

Need english editing
Author Response
Comments 1: Would you define the research objective ?
Response 1: Thank you for your helpful suggestion. In response, we have clarified and expanded lines 127–137 to more explicitly articulate our research aims and hypotheses. Specifically, we now highlight the investigation of OSA’s association with oral function, its variation across socioeconomic and health-related factors, and its potential disproportionate impact on vulnerable populations using nationally representative Korean data.
Comments 2: You need to provide more details on the sampling procedures and how did you evaluate and assess your variables?
Response 2: Thank you for your valuable suggestion. We have added further clarification on the sampling procedures and variable assessments in the revised manuscript. Specifically, we elaborated on how the Korea National Health and Nutrition Examination Survey (KNHANES) employs a multi-stage, stratified, cluster sampling design to ensure national representativeness.
Additionally, we provided more details on how key variables were measured and evaluated. For the assessment of obstructive sleep apnea (OSA) risk, we used the STOP-Bang questionnaire, which was administered through structured interviews conducted by trained personnel. To enhance transparency, we have included the full text of the STOP-Bang items and scoring criteria in the Supplementary Material (L189).
We appreciate your insight, which helped us strengthen the clarity and reproducibility of our Methods section.
Comments 3: Some figures need further editing with clear referencing
Figure 1: The figure lacks a clear citation in the main text. While it appears in the Methods section (Page 4), (e.g., "As shown in Figure 1, participants were excluded for...").
There are general issues with referencing of tables and figures e.g.,subgroup analysis on Page 9 mentions ORs but does not cite Table 4,
Also, ensure that every statistical result is linked to a specific table or figure.
For example: The stratified analysis (Table 4) revealed that low education amplified OSA’s impact on speaking discomfort etc...
Response 3: Thank you for your helpful suggestions regarding the clarity and referencing of tables and figures. In response, we have carefully revised the manuscript to ensure that all figures and tables are explicitly cited and appropriately integrated within the main text. Specifically:
- We have inserted a clear in-text citation of Figure 1 in the Methods section at Line 165, describing the participant selection process (“As shown in Figure 1, participants were excluded based on…”).
- The previously unreferenced subgroup analysis table has been revised into a forest plot and is now presented as Figure 2, which is clearly cited in the Results section at Line 293, with language such as: “Figure 2 illustrates the stratified analysis results, indicating a stronger association between OSA and speaking discomfort among participants with low educational attainment.”
- Furthermore, all statistical results mentioned in the main text are now explicitly linked to the corresponding table or figure to ensure clarity and consistency throughout the manuscript.
We appreciate the reviewer’s attention to these important presentation details, and believe these changes enhance the readability and scientific rigor of the manuscript.
Comments 4: Would you elaborate more on the statistical analysis ?
Response 4: Thank you for your comment regarding the statistical analysis. In response, we have elaborated on the analytical procedures in the Methods section. Specifically, we replaced the term "subgroup analysis" with the more appropriate term "stratified analysis" to reflect the statistical method more precisely. Stratified logistic regression models were conducted separately by education level and income quartile to explore potential variations across socioeconomically disadvantaged groups. These changes have been made to enhance transparency and rigor in the analytical approach.
Comments 5: The references need to be updated
Response 5: Thank you for your comment. We have reviewed and updated the references during the revision process to ensure the inclusion of more recent and relevant literature.
Comments 6: The limitations of the study are lacking
Respsonse 6: Thank you for your valuable feedback. We carefully reviewed the manuscript to identify and revise grammatical errors and awkward phrasing throughout the text. Accordingly, we have made a few modifications to improve the overall clarity, fluency, and readability of the manuscript. These changes were applied consistently across all sections, including the abstract, introduction, methods, results, and discussion. We hope that the revised version now meets the standards for language quality and is easier to read and interpret.
Please let us know if there are any specific sentences or sections that still require further revision.
Comments 7: Figures and tables need to be standardized
Table 2: The chi-squared test results are reported without highlighting key patterns (e.g., OSA high-risk groups showed significantly higher rates of chewing discomfort).
Table 4: The stratified analysis by income/education is noted, but the text does not summarize the most critical disparities (e.g., Low-income groups had 2x higher odds of chewing discomfort).
Response 7: Thank you for your valuable comment. In response, we revised the description of Table 2 results to highlight the key patterns in oral function problems across OSA risk groups. Specifically, we reported the percentage of each oral function issue by OSA risk level and included chi-squared test results to clearly demonstrate statistically significant differences (L260).
For Table 4, the original Table 3 has been replaced with a forest plot (Figure 2) to enhance clarity and visual interpretability of the adjusted associations. In this figure, individuals with higher OSA risk consistently exhibited greater likelihood of experiencing oral function problems, including chewing discomfort, speaking discomfort, and dental pain. Accordingly, we have revised to highlight the most critical disparities observed analysis. These additions provide a clearer summary of the most prominent socioeconomic disparities and align with your valuable suggestion (L292).
Reviewer 5 Report
Comments and Suggestions for Authors
Thank you for the submission of this interesting work.
This manuscript “The Impact of Obstructive Sleep Apnea (OSA) on Oral Function, using the Korea National Health and Nutrition Examination Survey (KNHANES) data” refers to the impact of OSA on oral function, and the enhancement of individual health and the broader public health landscape, ultimately advocating for policy support to improve health outcomes across society. OSA is strongly associated with an increased risk of developing serious secondary health conditions, including hypertension, diabetes, heart disease, and myocardial infarction. OSA and a spectrum of oral health issues, such as bruxism, xerostomia (dry mouth), periodontal disease, temporomandibular joint disorders, palatal and dental alterations, and even changes in taste sensation.
Through this research, the authors explore continuous management strategies for OSA, particularly in vulnerable populations, and to emphasize the risks associated.
The topic is within the journal scope, and it can help to understand the current state of knowledge about impact of Obstructive Sleep Apnea (OSA) on Oral Function.
Please insert the reference:
Sleep apnea is broadly classified into central and obstructive types. 45
Please rewrite these sentences:
he insidious nature of OSA, occurring 49
esity has been 57
Please rewrite Discussion section (228) and present advantages of this study, because you described its limitations.
Please rewrite Conclusions part (480): the authors should provide explanation how these findings could impact on public health interventions in this area.
In the Reference section 10 out of 37 references were within the last 5 years. It does not include excessive number of self-citations (3 references out of 37).
Author Response
Comments 1: Thank you for the submission of this interesting work.
This manuscript “The Impact of Obstructive Sleep Apnea (OSA) on Oral Function, using the Korea National Health and Nutrition Examination Survey (KNHANES) data” refers to the impact of OSA on oral function, and the enhancement of individual health and the broader public health landscape, ultimately advocating for policy support to improve health outcomes across society. OSA is strongly associated with an increased risk of developing serious secondary health conditions, including hypertension, diabetes, heart disease, and myocardial infarction. OSA and a spectrum of oral health issues, such as bruxism, xerostomia (dry mouth), periodontal disease, temporomandibular joint disorders, palatal and dental alterations, and even changes in taste sensation.
Through this research, the authors explore continuous management strategies for OSA, particularly in vulnerable populations, and to emphasize the risks associated.
The topic is within the journal scope, and it can help to understand the current state of knowledge about impact of Obstructive Sleep Apnea (OSA) on Oral Function.
Response 1: We sincerely thank the reviewer for their thoughtful and encouraging feedback. We appreciate your recognition of the study's relevance and contribution to advancing the understanding of the relationship between OSA and oral function, especially in the context of public health. Your comments motivate us to further refine our work and enhance its academic and practical value.
Comments 2: Please insert the reference:
Sleep apnea is broadly classified into central and obstructive types. 45
Please rewrite these sentences:
he insidious nature of OSA, occurring 49
esity has been 57
Response 2: Thank you for your careful reading. We have inserted the appropriate reference to support the sentence “Sleep apnea is broadly classified into central and obstructive types” as Reference [45].
Additionally, we have revised the following typographical or unclear phrases for clarity:
- Line 49: “The insidious nature of OSA, occurring during sleep, often leaves individuals unaware of their condition…” (corrected grammar and clarity)
- Line 57: “Obesity has been identified as a major risk factor for OSA…” (corrected spelling from “esity” to “obesity”).
Comments 3: Please rewrite Discussion section (228) and present advantages of this study, because you described its limitations.
Response 3: Thank you for this helpful suggestion. In response, we have revised the Discussion section to better highlight the strengths of our study(L494-509). These revisions aim to provide a more balanced discussion by complementing the limitations with the study’s methodological and public health strengths.
Comments 4: Please rewrite Conclusions part (480): the authors should provide explanation how these findings could impact on public health interventions in this area.
Response 4: Thank you for your thoughtful revision. The Conclusion section has been revised to include implications for public health (L530-542). We now state that our findings underscore the need for integrated OSA screening and oral function assessment, especially among socioeconomically vulnerable populations. We also suggest that community-based programs combining sleep health education and oral health promotion may help reduce disparities in both areas. These revisions aim to clearly present how the study findings could inform future health interventions and policy development.
Comments 5: In the Reference section 10 out of 37 references were within the last 5 years. It does not include excessive number of self-citations (3 references out of 37).
Response 5: Thank you for your assessment. We appreciate your acknowledgement regarding the appropriate use of self-citations. During the revision process, we have added several recent studies (within the last 5 years) in response to reviewer suggestions and to strengthen the literature foundation. We have been mindful to ensure that these additions improve the currency and relevance of the citations while avoiding citation inflation.
Round 2
Reviewer 1 Report
Comments and Suggestions for Authors
Thank you for the comments.
Please, consider only adding the information on previous validation of the study. - another paper was mentioned by you in an explanation letter. (this is important, as it might differ between countries).
It does not influence my decission on endorsing the publication anyway.
Thank you.